# One- and Two-Year Effects of the Healthy Primary School of the Future on Children’s Dietary and Physical Activity Behaviours: A Quasi-Experimental Study

**DOI:** 10.3390/nu11030689

**Published:** 2019-03-22

**Authors:** Nina H. M. Bartelink, Patricia van Assema, Stef P. J. Kremers, Hans H. C. M. Savelberg, Marije Oosterhoff, Maartje Willeboordse, Onno C. P. van Schayck, Bjorn Winkens, Maria W. J. Jansen

**Affiliations:** 1Department of Health Promotion, Care and Public Health Research Institute (CAPHRI), Maastricht University, P.O. Box 616, 6200 MD Maastricht, The Netherlands; p.vanassema@maastrichtuniversity.nl; 2Department of Health Promotion, School of Nutrition and Translational Research in Metabolism (NUTRIM), Maastricht University, P.O. Box 616, 6200 MD Maastricht, The Netherlands; s.kremers@maastrichtuniversity.nl; 3Academic Collaborative Centre for Public Health Limburg, Public Health Services, P.O. Box 33, 6400 AA Heerlen, The Netherlands; maria.jansen@ggdzl.nl; 4Department of Nutrition and Movement Sciences, Nutrition and Translational Research in Metabolism (NUTRIM), Maastricht University, P.O. Box 616, 6200 MD Maastricht, The Netherlands; hans.savelberg@maastrichtuniversity.nl; 5Department of Clinical Epidemiology and Medical Technology Assessment (KEMTA), Maastricht University Medical Centre MUMC+/ Care and Public Health Research Institute (CAPHRI), Maastricht University, P.O. Box 5800, 6202 AZ Maastricht, The Netherlands.; marije.oosterhoff@mumc.nl; 6Department of Family Medicine, Care and Public Health Research Institute (CAPHRI), Maastricht University, P.O. Box 616, 6200 MD Maastricht, The Netherlands.; maartje.willeboordse@maastrichtuniversity.nl (M.W.); onno.vanschayck@maastrichtuniversity.nl (O.C.P.v.S.); 7Department of Methodology and Statistics, Care and Public Health Research Institute (CAPHRI), Maastricht University, P.O. Box 616, 6200 MD Maastricht, The Netherlands; bjorn.winkens@maastrichtuniversity.nl; 8Department of Health Services Research, Care and Public Health Research Institute (CAPHRI), Maastricht University, P.O. Box 616, 6200 MD Maastricht, The Netherlands

**Keywords:** behaviour, nutrition, physical activity, effect evaluation, health-promoting schools, follow-up

## Abstract

Schools can help to improve children’s health. The ‘Healthy Primary School of the Future’ (HPSF) aims to sustainably integrate health and well-being into the school system. This study examined the effects of HPSF on children’s dietary and physical activity (PA) behaviours after 1 and 2 years’ follow-up. The study (*n* = 1676 children) has a quasi-experimental design with four intervention schools, i.e., two full HPSF (focus: nutrition and PA) and two partial HPSF (focus: PA), and four control schools. Accelerometers and child- and parent-reported questionnaires were used at baseline, after 1 (T1) and 2 (T2) years. Mixed-model analyses showed significant favourable effects for the full HPSF versus control schools for, among others, school water consumption (effect size (ES) = 1.03 (T1), 1.14 (T2)), lunch intake of vegetables (odds ratio (OR) = 3.17 (T1), 4.39 (T2)) and dairy products (OR = 4.43 (T1), 4.52 (T2)), sedentary time (ES = −0.23 (T2)) and light PA (ES = 0.22 (T2)). Almost no significant favourable effects were found for partial HPSF compared to control schools. We conclude that the full HPSF is effective in promoting children’s health behaviours at T1 and T2 compared with control schools. Focusing on both nutrition and PA components seems to be more effective in promoting healthy behaviours than focusing exclusively on PA.

## 1. Introduction

Dietary and physical activity (PA) habits are formed at a young age [1], whereby unhealthy habits can already lead to overweight and obesity [2]. The health behaviours of children are suboptimal in Western countries, including the Netherlands: 42% of children (aged 4–9 years) consume at least 150 g of fruit per day, this percentage drops to 20% for 9–12 year olds. The prevalence of vegetable intake shows similar percentages: 41% of 4–9 year olds and 25% of 9–12 year olds eats at least 150 g of vegetables per day [3]. Regarding PA, only half (48%) of Dutch children (aged 4–12) meet the guidelines for PA of 60 min of moderate-to-vigorous physical activity (MVPA) per day [4]. Consequently, 13–15% of Dutch children (aged 2–21 years) are overweight, and 1.8–2.2% are classified as obese, which is a two- to three-fold increase compared with 1980 [5]. Childhood overweight often tracks into adulthood [6] and is related to health and psychosocial problems, reduced quality of life, and higher health care costs [7,8,9]. An association also exists between health and educational achievement: Health status affects the capacity to learn, while educational achievements affect health status [10]. This link between health and education often results in persistent socioeconomic health inequity problems that continue to exist from generation to generation [11,12]. 

Schools are increasingly recognized as significant in improving children’s health behaviours since a large proportion of a child’s day is spent there, and schools reach all children [13,14]. However, school-based health interventions are often not integrated in the school system and are characterised by relatively low priority, a lack of coordination, and are often supply-driven, resulting in limited effects or effects that diminish in the long term [15,16]. The Health-Promoting School (HPS) framework, initiated by the World Health Organization, aims for a whole-school approach, with a focus on reorienting school systems toward sustainable health promotion [17]. HPS focuses not only on classroom-based health education, but also on changes in school policy and the schools’ physical and social environment, using bottom-up involvement of pupils, parents, teachers and staff. Several reviews have been published on the effectiveness of HPS in improving the health and well-being of school children [18,19,20,21]. Even though the findings indicate small favourable effects in terms of PA and healthier food choices, the reviews also reveal that the findings were not uniform across the included studies. Many studies showed suboptimal results, often due to a short duration of the intervention, a lack of a whole-school approach and implementation challenges [22,23,24]. Implementation challenges can be considered a result of the interaction between the intervention and the specific context [25,26,27]. Therefore, various studies suggest revising the idea of interventions as something fixed or static, and considering them as ‘events’ occurring within the school system [28]. 

The ‘Healthy Primary School of the Future’ (HPSF) is a Dutch initiative based on the HPS framework (including, e.g., whole school approach, participation, partnerships) and embraces the contextual systems approach [29,30]. This initiative aims to sustainably integrate health and well-being within the school system. 

The processes and effects of the initiative, implemented in four pilot schools, are being investigated in an overall study by a multi-disciplinary research group [29,30]. The primary outcome of the overall study is children’s BMI *z*-score, which significantly decreased after 2 years’ follow-up in the HPSF schools compared to control schools [31]. The current study focuses on two key aspects of HPSF, i.e., healthy nutrition and PA. Recent research suggests that by addressing two clustered health behaviours, a spill-over or synergistic effect might occur, whereby the probability of enhancing one health behaviour increases when an individual has successfully changed the other health behaviour [32,33]. This means that, for example, an increase in physical activity may lead to improved eating behaviours and vice versa. Therefore, simultaneously addressing healthy nutrition and PA might be more effective due to the facilitation of this potential synergistic effect. 

The aim of the current study is to examine the effects of HPSF on children’s dietary and PA behaviours after 1 and 2 years’ follow-up compared with control schools, with two schools focussing on both nutrition and PA (full HPSF), and two schools focussing only on PA (partial HPSF). We hypothesized that in the full HPSF, effects will be noted on both dietary and PA behaviours, and in the partial HPSF mainly on PA behaviours. Additionally, we hypothesized that larger effects will be found in the full HPSF, due to the potential synergy between dietary and PA behaviours in children. 

## 2. Materials and Methods

### 2.1. Study Design

The current study has a longitudinal quasi-experimental design with four intervention schools (two full and two partial) and four control schools, which maintained the school curriculum that is currently common practice in the Netherlands [30]. Inclusion criteria for schools include being a member of the educational board ‘Movare’, since they were one of the initiators of HPSF, and a minimum of 140 children in the study years two till five, to be able to study the effects of HPSF with enough power. The schools are all situated in the Parkstad region in the southern part of the Netherlands. This region has a low average socio-economic status (SES), and unhealthy behaviours and overweight are highly prevalent compared with the rest of the Netherlands [34,35]. Ethical approval (14-N-142) for the overall study was obtained from the Medical Ethics Committee Zuyderland located in Heerlen (Parkstad, the Netherlands). All participants were required to complete an informed consent form, signed by (both) parents. All four intervention schools started implementation of HPSF in November 2015. Funding for implementation is provided until the end of 2019. However, the four schools have committed to continued implementation after 2019 and make the changes sustainable in their school. Measurements in all eight included schools were conducted in September–November of 2015 (T0), 2016 (T1) and 2017 (T2); the overall study continues until 2019. The data that support the findings of this study were collected as part of the ‘Healthy Primary School of the Future’ quasi-experimental study. Data collection will take place until 2019 to study the effects after 4 years of exposure. Following article publication, data will become available on the 4-year effects and other potentially comparative studies in the Netherlands. A detailed description of the overall study and the recruitment of the schools is reported in Willeboordse et al. [30]. The study was retrospectively registered in the ClinicalTrials.gov database on 14 June 2016 (NCT02800616). 

### 2.2. The Healthy Primary School of the Future

Three collaborating organisations, i.e., the regional educational board ‘Movare’, the regional Public Health Services and Maastricht University, developed the HPSF initiative. The initiative is based on the principles of the HPS framework and aims to sustainably integrate health and well-being within the whole school system. To achieve this aim, HPSF intends to establish a broad collaboration between school, parents, and external partners, which should lead to a co-creation movement in schools. This includes top-down and bottom-up processes to develop and implement together health-promoting changes in all aspects of the school system, e.g., school’s physical and social environment, school’s health policy, education, and school routines. This also refers to sustainability research, whereby, among other things, partnership, ownership, organizational routines, and add-in changes, are important factors for success [36,37,38]. 

On top of the HPS framework, the aim was to create some form of positive disruption in the school, by initiating two changes top-down: (1) a free healthy lunch each day (only in full HPSF) and (2) structured PA sessions after lunch. These changes are contextualized bottom-up and should lead to momentum for bottom-up processes to institutionalise health-promoting routines in the school. The time for having lunch (in the full HPSF) was increased to 20–30 min. The total lunch break time in these schools was prolonged by about 60 min. For this reason, the school day was extended: Children of the full HPSF attend school to approximately 15:30/15:45 instead of 15:00. A dietician of the caterer developed a lunch menu cycle that changed every 10 weeks, in which at least 80% of the products met the advice of the Dutch Health Council [39]. A mid-morning snack, consisting of fruits and/or nuts, was also provided. The lunch, a bread-based cold meal, was typically Dutch. During lunch break time, the children participated several times a week in structured PA sessions; one or two times per week they could participate in cultural activities. The PA sessions were carried out in the schoolyard and, when available and needed, in parks, forest, and/or sports hall in the neighbourhood. All schools collaborated with sport clubs or other external partners to offer specific activities as well. Since the two changes were contextualized bottom-up, this resulted in some differences between schools in the form of the changes; the content remained comparable. To not increase the workload of teachers even further, the top-down changes were implemented by external pedagogical employees provided by childcare organizations. This integration of the childcare organization during school hours is not to provide a temporary solution, but to change the school’s organization in a sustainable way. The aim for the future is to bring school and childcare more together and thereby create an integrated day for children, whereby children are supervised by the same people prior, during and after school hours. The abovementioned commitment of schools and childcare organizations to continued implementation, also includes this employment of external pedagogical employees during school hours. Employees of sports and leisure organizations supported the pedagogical employees during implementation when needed, and after a year they provided a training course (8 sessions of 2 h) to supply them with additional tools for how to motivate children for active participation during the PA sessions. 

The implementation of the lunch and the duration of the lunch break time were the main differences between the two versions of HPSF. The full and partial HPSF implemented the structured PA sessions in a comparable way and had quite similar support from external partners [40]. Both the full- and partial-HPSF schools involved teachers and parents in the adoption decision and the process of adapting the two changes into the school context. All four intervention schools used a children voice group, with representatives from each class in school, to get insight into the opinion of children regarding HPSF. In this way, the experiences of children were being heard and the changes could be further contextualized to fit better to the children’s needs and wishes. Differences existed in the implemented additional health-promoting changes [40]. The full HPSF improved their health policy, provided water bottles to all children, and provided an educational lunch once a week. The partial HPSF did not implement additional health-promoting changes. Each of the four intervention schools selected a teacher as school coordinator, who managed HPSF in their school. Overarching the four schools, the HPSF initiative was led by a project leader from Movare and an executive board with representatives of the three collaborating organisations, including the project leader. A project team was created with representatives of all partners involved: the four schools, Movare, regional Public Health Services, Maastricht University, the Limburg provincial authorities, childcare organizations, the caterer, and sports and leisure organizations. 

### 2.3. Study Population

All children (age 4 to 12) and their parents from the eight schools (*n* = 2326 at T0) were invited to participate in the study; no inclusion or exclusion criteria were set. This included children from study year one to eight, which is comparable to 2 years of Kindergarten and six grades. Recruitment was done via information brochures for parents. In addition, the research team visited the classrooms to inform children about the study and encourage them to ask their parents to participate. Due to the dynamic population in the schools (new children enter and other children finish school each year), we focused in this study only on the children who were enrolled in the schools at baseline till the end of this 2-year study. In this way, only children were included in the current study who participated in the full 2 years of HPSF in their school. The group of children included in this study were: At baseline (T0), children from study year one to seven; at T1, children from study year two to eight; and at T2, children from study year three to eight. Children who joined the study at T1 or T2 were included, even though no baseline data were available. Even though these children joined the study after 1 or 2 years, they were at baseline already participating in their school and thus also exposed to HPSF during the full 2 years of this study. Children who switched to other schools between T0 and T2 were excluded. 

### 2.4. Data Collection Procedures

In each school, the data were gathered during 1 week of measurements. Inter-rater variability was minimised by training researchers according to a strict protocol. 

#### 2.4.1. Accelerometers

At the beginning of the measurement week, all participating children from study year two to eight received an accelerometer for 7 days (Actigraph GT3X+, 30 Hz, 10 s epoch). The monitor was attached to the hip with an elastic band and had to be worn all day except while sleeping or during activities in which water was involved (e.g., swimming, bathing and showering). To control for the influence of weather on PA levels, data on weather conditions between 6 a.m. and 11 p.m., e.g., mean temperature, sun exposure and precipitation, were collected from the Royal Dutch Meteorological Institute (KNMI). 

#### 2.4.2. Questionnaires

Children and one of their parents were asked to fill out one (parents) or two (children) questionnaires. The child questionnaires were based on the validated parent questionnaire, but simplified to make it appropriate for children. We did not validate these adapted questionnaires. However, all questionnaires were pretested, for difficulty, length and content by experts in the field of health promotion, the target group, e.g., individual children and parents, and classes of children in a primary school. 

*Parent questionnaire:* A digital questionnaire for parents was used to obtain information about, among others, the education level and country of birth of both parents, household income, and children’s health behaviours. To assess children’s PA behaviours, 14 questions were used from the Local and National Youth Health Monitor [41]. These questions in the monitor were based on the International Physical Activity Questionnaire, which has acceptable validity [42]. Parents were asked how many days a week and how many minutes a day their child engaged in several PA activities (e.g., active transport, leisure time PA indoors and outdoors, and sports clubs) and sedentary activities (e.g., watching television, computer use and social media use) during the past week. Twelve questions from the Local and National Youth Health Monitor were used to assess children’s dietary behaviours [43]. These questions were based on the short Fat List, which has acceptable validity [44]. Parents were asked about the number of days (on a scale from 0–7 days a week) their child consumed breakfast; ate warm vegetables, salads or raw vegetables, and fruits; and consumed water and sugar-sweetened beverages (soft-, sports-, and energy-drinks) during the past week. They were also asked how many times a week their child ate the following four snack types: chocolate, salted snacks, cookies and soft ice-creams (on a scale of 0–7 days a week). All parents of participating children (study years one to eight) received the questionnaire. It took about 60 min for the parents to fill in as other aspects were also explored, such as quality of life. Parents had approximately 1–3 months to fill in the questionnaire: From the start of each measurement week until the end of the calendar year. Two reminders were sent in this period if the questionnaire was not yet completed. 

*Child questionnaire:* The questionnaire was filled in by children of study years four to eight and was used to assess their dietary behaviours and their water consumption specifically in school. Questions regarding the children’s PA behaviours mainly focused on whether they liked specific activities and were not used in the current study. Twelve questions were included, based on the Local and National Youth Health Monitor, regarding daily breakfast intake, the intake of fruit and vegetables, the consumption of water (at school), sugar-sweetened beverages (soft-, sports-, and energy-drinks), and the consumption of the four snack types (chocolate, salted snacks, cookies and soft ice-creams) [43]. The reply options were simplified to (1) never or almost never, (2) sometimes (1–3 days per week), (3) often (4–6 days per week), and (4) every day; the reply for daily breakfast intake was yes/no. The questionnaire was filled out by hand during class hours in the presence of at least one member of the research team. It took about 40 min to fill out, as other aspects such as quality of life were also included. 

*Child lunch questionnaire:* The questionnaire regarding children’s lunch intake was filled out by children of study years three to eight and consisted of recall questions (*n* = 8) with yes/no reply options regarding the consumption of food types that are included in the Wheel of Five designed by The Netherlands Nutrition Centre [45], i.e., bread, cereals, butter, cheeses, fruits, vegetables, milk/yoghurt, and water during lunch that day. The questionnaire was filled out by hand immediately after lunch time, which took about 5 min. 

## 3. Measures

### 3.1. Covariates

Children’s gender, age and study year at baseline were collected via the database of the educational board Movare. SES was calculated as the mean of standardized scores on maternal education level, paternal educational level, and household income (adjusted for household size) [46]. SES scores were categorized into low, middle and high based on tertiles. Children’s ethnicity was determined by the country of birth of both parents and divided into (1) Western background (including the Netherlands) and (2) non-Western background [47]. If one of the parents was born in a non-Western country, the child’s ethnicity was assigned to non-Western. Body Mass Index (BMI) was assessed by anthropometric measurements of height and weight [30]. BMI z-scores were calculated using Dutch reference values [5]. 

### 3.2. Outcomes

*Children’s PA behaviours:* PA levels derived from the accelerometry data were processed using ActiLife version 6.13.3 (ActiGraph, Pensacola, FL, USA). Wear time validation was assessed using Choi’s classification criteria [48]. Minimal wear time was defined as 480 min per day between 6 a.m. and 11 p.m. [49]. The first day of measurement was excluded to prevent reactivity [50]. Measurements containing at least three weekdays (after excluding the first measurement day) and one weekend day were used in the analyses [51]. Mean temperature, sun exposure and precipitation were merged with the accelerometry data to obtain weather scores for all days the child wore the accelerometer. The activity levels in counts-per-minute (CPM) were classified using Evenson’s cut-off points [52]: sedentary behaviour (SB; ≤100 CPM), light PA (LPA; 101–2295 CPM), and moderate to vigorous PA (MVPA; ≥2296 CPM). The children’s total time spent on PA and sedentary behaviours was derived from the parent questionnaire. The number of days per PA behaviour (active transport, leisure time PA inside and outside, and sport clubs) or sedentary behaviour (watching TV, using computer, social media use) were multiplied by the average number of minutes spent in a day and divided by seven (active transport was divided by five). The four specific PA behaviours were summed into a PA behaviours total score, and the three sedentary behaviours were summed into a sedentary behaviours total score. Missing values were imputed using a child’s mean imputation if there were not too many items missing (scales with <5 items: max 1 item missing; scales with ≥5 items: max 2 items missing), otherwise they were considered as missing PA behaviour. 

*Children’s dietary behaviours:* total scores for healthy and unhealthy dietary behaviours were used, due to the high number of dietary outcomes, and the fact that small changes in several specific dietary behaviours could be better detected by using total scores. A total score for healthy behaviours was calculated by the mean number of days (parent-reported) and the mean score (child-reported) of breakfast consumption, intake of fruits, vegetables (parent-reported: distinction between warm and cold) and water. A total score for unhealthy behaviours was calculated by the mean number of days (parents) or mean score (child) of intake of sugar-sweetened beverages and the four different snack types. To be able to include breakfast intake in the child-reported total score, this score had to be recoded to (1) not every day and (3) every day. Missing values were imputed using a child’s mean imputation if there were not too many items missing (<5 items: max 1 missing; ≥5 items: max 2 missing). The variable ‘school water consumption’ (range: 0 (never)–3 (every day)) of the child questionnaire was used to assess children’s water intake in school in particular. 

*Children’s lunch intake:* The following six food types were derived from the child lunch questionnaire: fruits, vegetables, grains, dairy, water and butter. The items bread and cereals were combined into the food type grains, and milk/yoghurt and cheese were combined into the food type dairy. To give an indication of the nutritional value of children’s lunch, we summed the six different food types consumed and created a dichotomous variable to study whether children consumed at least two of the food types during lunch. Additionally, to investigate if change occurred in the consumption of specific combinations of food types, we created five variables for the most common combinations, e.g., grains and fruit, grains and vegetables, dairy and fruit, dairy and vegetables, and grains and dairy. 

## 4. Statistical Analyses

Data were analyzed using IBM SPSS Statistics for Windows (version 23.0, IBM Corp, Armonk, NY, USA). Pearson’s chi-square tests and ANOVA tests were conducted to analyze the comparability of observed participant characteristics at baseline, i.e., gender, study year, SES status, ethnicity, BMI *z*-score, and PA and dietary behaviours, among the full HPSF, the partial HPSF, and control schools. The percentage of children who improved in a specific behaviour after 1 and 2 years, i.e., changed in a favourable direction compared with their baseline result, was studied by descriptive statistics. Linear mixed model analyses were used to assess the longitudinal intervention effects on children’s PA levels and behavioural outcomes; Generalized Estimating Equations were used for binary outcomes. Since measurements were repeated, within participants we used a two-level model with measurements as the first level and participants as the second level. The fixed part of the model consisted of group (full HPSF, partial HPSF, control), time (T0, T1, T2) and the interaction terms of group with time. We were not able to include class as a level in the model, because commonly more than one division of a class existed, e.g., 4a or 4b, and children often did not have fixed class divisions for all years. All analyses were adjusted for the covariates: gender, study year at baseline, SES, ethnicity, and children’s BMI *z*-score at baseline. The analyses regarding children’s PA levels were also adjusted for weather conditions (mean temperature, sun exposure in hours/day and precipitation in hours/day). Missing data, including missing data at baseline, were imputed using a multiple imputation method with fully conditional specification (FCS) and 10 iterations, generating 50 complete datasets. BMI *z*-score, gender, study year at baseline, school type, ethnicity, SES score, temperature, sun exposure, and precipitation were used to obtain a complete covariate set, with a likelihood-based approach being used for missing outcome variables. This latter was done for practical reasons as the number of outcome variables was too large. A two-sided *p*-value ≤ 0.05 was considered statistically significant. Standardized effect sizes (ES) were determined for numerical outcomes, which were computed as pooled estimated mean difference divided by the square root of the pooled residual variance at baseline. Binary outcomes resulted in odds ratios. 

## 5. Results

At baseline (T0), 2326 children and their parents were invited to participate in the overall study to investigate the effects of HPSF; 60.3% joined the study (*n* = 1403). Because of the study’s dynamic population, a total of 1974 children and their parents participated in the study within the 2-year follow-up period (data collected at one time-point at least). Due to the selection used for the current study, i.e., only including the children who were in study years one to seven at baseline and excluding school switchers, we included 1676 children in the analyses. This selection and the study’s flow diagram are similar to the study that investigated the 1- and 2-year effects of HPSF on children’s BMI *z*-score [31]. Of these children, 47.4% were boys, the mean age was 7.5 years old, and 94.1% had a Western ethnicity. In the full HPSF, 537 children were included, in the partial HPSF, 478 children, and in the control schools, 661 children. No covariates differed significantly at baseline between the three school groups, except for BMI *z*-scores (*p* = 0.034): children in the control schools (BMI-*z* = 0.232) had a higher mean BMI *z*-score compared with children of the full HPSF (BMI-*z* = 0.051) and the partial HPSF (BMI-*z* = 0.092). Regarding children’s dietary and PA behaviours, many significant differences existed at baseline, with unhealthier behaviours mostly found in the children in the control schools compared to the full and partial HPSF (Table 1, Table 2 and Table 3). Not all parents filled out the parent questionnaire: Parents of 1115 children (66.5%) completed the questionnaire at least once. The child questionnaire was filled out at least once by 96.1% of the children, the child lunch questionnaire by 98.3% of the children. Sufficient accelerometer data, i.e., enough wear time to be included in the analyses, in at least one measurement was reached in 81.5% of the children. 

### 5.1. Children’s PA Behaviours

Significant favourable intervention effects were found in the accelerometry data in the full HPSF versus control schools (Table 1). The percentage time spent sedentary had decreased more (ES = −0.23) and the percentage time spent in light PA had increased more (ES = 0.22) at T2 in children of the full HPSF compared with control schools. More than a quarter of all children (28.2%) improved, i.e., decreased their sedentary time at T2 in the full HPSF, which was more than the percentage of children in the control schools (21.6%). The percentage time spent in MVPA did not differ significantly in the full HPSF compared with control schools. However, the percentage of children who improved their time spent in MVPA was higher in the full HPSF (44.4%) than the control schools (35.8%). The parent-reported data regarding children’s PA behaviours showed mixed results: The total time per day spent on both PA behaviours (ES = −0.22) and sedentary behaviours (ES = −0.29) had decreased more at T2 in the full HPSF compared with control schools. In the partial HPSF, no significant intervention effects were found in the accelerometry data or parent-reported data compared with control schools (ES between −0.07 and 0.08). 

### 5.2. Children’s Dietary Behaviours

Significant favourable intervention effects were found for parent-reported children’s dietary behaviours in the full HPSF. Children’s healthy dietary behaviours (total score for breakfast, fruit, vegetables, and water) improved significantly more in the full HPSF compared with control schools at T1 (ES = 0.20) and T2 (ES = 0.19) (Table 2). Effect sizes per item of this total score were largest for water consumption (Appendix A). Children’s unhealthy dietary behaviours decreased significantly more for the full HPSF versus control schools at T1 (ES = −0.23). A significant favourable intervention effect was also found for child-reported water consumption at school: at T1 and T2, a significantly higher increase was found in children of the full HPSF compared with control schools (T1: ES = 1.03; T2: ES = 1.14). More than three-quarters of all children improved, i.e., increased their water consumption at school at T1 and T2 in the full HPSF, which was almost double the percentage of children compared with the control schools. In the partial HPSF, no significant intervention effects were found for parent-reported children’s dietary behaviours compared with control schools (ES between −0.14 and 0.07). Results on child-reported unhealthy dietary behaviours showed a significant favourable intervention effect at T2: a significantly larger decrease in the partial HPSF compared with control schools (ES = −0.25). 

### 5.3. Children’s Lunch Intake

Significant intervention effects were found for children’s lunch intake (child-reported) in the full HPSF: A significantly higher increase was found at T1 for the consumption of fruit (OR = 2.63), vegetables (OR = 3.17) and dairy products (OR = 4.43) compared with control schools (Table 3). These higher increases remained significant at T2 for the consumption of vegetables (OR = 4.39) and dairy products (OR = 4.52). The consumption of grains and butter during lunch decreased significantly more at T1 (grains: OR = 0.43; butter: OR = 0.22) and T2 (grains: OR = 0.45; butter: OR = 0.19) in the full HPSF compared with control schools. The consumption of at least two food types during lunch increased significantly more in the full HPSF compared with control schools (OR = 3.51 (T1) and 2.98 (T2)). The consumption of five common food type combinations improved by approximately 30–40% at T1 and T2 in the full HPSF. In contrast, this percentage was much less in the control schools (8–20%) (Appendix A). In the partial HPSF, the consumption of vegetables (OR = 0.58), dairy products (OR = 0.45) and butter (OR = 0.64) during lunch significantly decreased more at T2 compared with control schools. 

## 6. Discussion

HPSF is a health-promoting school initiative that uses a contextual systems approach [17,29,30]. The initiative aims to create health-promoting changes in different aspects of the school system, i.e., school’s physical and social environment, school’s health policy, education, and school routines. On top of the HPS framework, the aim was to create some form of positive disruption in the school, which should lead to momentum for bottom-up processes to institutionalise health-promoting routines in the school. The aim of the current study was to examine the effects of HPSF on children’s dietary and PA behaviours after 1 and 2 years’ follow-up compared with control schools. Favourable intervention effects on children’s dietary and PA behaviours were found for the full HPSF. In contrast, almost no significant favourable results were found for the partial HPSF, where we expected favourable effects on children’s PA behaviours. These effects are in line with the findings of the review of Langford et al., who investigated comparable school-based initiatives [18]. This review stated as well that PA behaviours significantly improved only in the initiatives with a focus on both healthy nutrition and PA behaviours, and not in initiatives that focused solely on PA behaviours. In contrast to our study, this review did not find any significant results on behaviours related to healthy nutrition for the schools with a focus on both healthy nutrition and PA. However, comparison is limited, since we used total scores and this review used fruit and vegetable intake and fat intake as outcomes. 

The results found in the full HPSF regarding the accelerometry data can be seen as small intervention effects according to Lipsey’s guidelines for effect sizes (small (0–0.32), medium (0.33–0.55) and large (>0.55)) [53]. The significant effects on children’s sedentary time and light PA, but not on MVPA, are in line with other studies of school-based initiatives [54,55]. Contrary to the accelerometry-data, parent-reported data showed in the full HPSF not only a favourable effect (decrease in sedentary behaviours), but also an adverse effect (decrease in PA behaviours). These differences in effects found by using the two methods might be explained because assessing PA by subjective parent-reported questionnaires has a lower validity than objective accelerometry [56]. However, the differences might also be due to the focus of the PA-related questions for parents being outside of school hours, while the accelerometers assessed PA over the whole day. Children of the full HPSF have less time for sedentary and PA behaviours outside of school because of the extended school day. Since both behaviours decreased, it does not necessarily mean that the extra PA at school resulted in children compensating for PA outside of school hours, which has been found in other studies [57]. More in-depth research is needed to investigate the difference in effect during and outside of school hours on children’s PA behaviours. 

The large favourable intervention effect on school water consumption in the full HPSF is probably a result of implementing additional health-promoting changes related to water, e.g., handed out water bottles to all children and improved their school water policy, which created a more health-promoting environment and policy in the school. Both are important aspects of the HPS framework. Both schools referred to the momentum in the school created by the lunch to implement these water-related changes [40]. 

Furthermore, the increase in the consumption of fruits, vegetables and dairy products, and the decrease in grains and butter in the full HPSF seem to indicate that children eat more different food types during lunch. Their lunch intake seems to have changed from a typical Dutch bread-based lunch to a more diverse lunch. The large favourable intervention effect on the intake of at least two food types during lunch seems to validate this conclusion. 

The intervention effects in the full HPSF were quite similar at both time points, and the T2 intervention effects were even higher for children’s PA behaviours than the T1 intervention effects. This seems to indicate that the effects are not only due to the children’s enthusiasm for and cooperation with the new changes in school, but that new habits and routines may have developed in the children’s health behaviours. However, longer follow-up periods are needed to investigate the long-term effects. 

The main difference between the full and partial HPSF was the implementation of the lunch, the duration of the lunch break time, and the implementation of additional health-promoting changes. However, the two versions of HPSF also had many similarities: both implemented the structured PA sessions in a comparable way, and they were quite similar in the coordination of HPSF and the support of external partners. Nonetheless, the full HPSF was more effective than the partial HPSF, also regarding children’s PA behaviours. Three possible explanations can be given. First, as hypothesized, simultaneously addressing nutrition and PA seemed to create a synergistic effect that led to greater effectiveness. Various studies have indeed suggested that dietary and PA behaviours are associated and that the probability of enhancing a second behaviour, e.g., PA, increases when an individual has successfully changed a first behaviour, e.g., healthy nutrition [32,33]. Second, both the full and partial HPSF used a contextual systems approach and included top-down and bottom-up processes to create health-promoting changes in the school [29]. Since the two top-down changes were also contextualized bottom-up, this resulted in some differences between schools in the form of the changes, e.g., assigning external pedagogical employees to a specific activity or to a specific class. The content of the changes remained comparable, however. Moreover, the results of the process evaluation of Bartelink et al. indicated that the lunch turned out to be a positive disruption in the full HPSF that created momentum for more bottom-up processes, including more involvement and support of teachers and parents, and it has led to additional health-promoting changes (e.g., health-promoting policy) [40]. The partial HPSF did not implement the lunch, which resulted in limited bottom-up processes and no additional health-promoting changes in these schools. Due to this lack of additional changes, the whole school approach as suggested by the HPS framework is limited, which might explain the differences in effect between the full and partial HPSF. Third, the partial HPSF did not extend the lunch break time, whereas the full HPSF created a longer break by extending it by approximately 60 min. Consequently, the time for the structured PA sessions was longer in the full HPSF compared with the partial HPSF. 

Although we hypothesized that differences in effect would exist between the full and partial HPSF, we did not expect that in the partial HPSF no effects on children’s PA behaviours would be found at all. An explanation for this absence might be that children’s PA behaviours in the specific weeks of measurements were not representative of the children’s PA behaviours in general. Moreover, the effects on children’s PA behaviours might also be too small to detect. The results of the study regarding the effects of HPSF on children’s BMI *z*-score found a small but significant decrease in BMI *z*-score after 2 years in both the full (ES = −0.08) and partial (ES = −0.07) HPSF [31], which suggests that also in the partial HPSF, some changes have occurred in the children’s health behaviours [58]. Many small improvements on several different health behaviours can lead to a decrease in BMI *z*-score, since it is the co-existence and interaction of specific nutrition and PA behaviours that results in a positive (or negative) energy balance and weight gain (or loss) [59,60]. 

## 7. Limitations and Strengths

The longitudinal quasi-experimental design can be seen as a limitation of this study, since we were unable to (cluster-) randomize schools. However, due to this design, we were able to test the effectiveness in terms of differences in children’s health behaviours between the three school groups over time, and were also able to enroll schools on the basis of motivation, which reflects the real-life situation of school health promotion. However, due to no randomization, it has probably resulted in significant baseline differences between the three groups. The baseline differences in BMI *z*-scores and health behaviours seem to indicate that children in the control schools have developed stronger habits in unhealthy behaviours, which have already led to more overweight or obesity. These stronger habits can be more difficult to change, but also show more room for improvement for the children in the control schools compared with the full and partial HPSF, which can result in an underestimation of the effects. To deal with the limitation of no randomization, we controlled in all analyses for BMI *z*-score at T0, gender, study year at T0, SES score, and ethnicity. Moreover, a methodological strength of the study is the objectively measured PA levels, all collected in the same season, and the matching of all measurements in the same week. 

In addition to the abovementioned methodological limitation regarding assessing PA behaviours among parents, the use of questionnaires in general had its limitations as these are subjective measurements, which may lead to socially desirable answers [61]. Therefore, we used different data sources to obtain information about the children’s health behaviours. The advantage of using questionnaires for children was that a high response rate (child questionnaire: 96.1%, lunch questionnaire: 98.3%) could be achieved by classical inquiry; children are often more honest in their answers and children’s behaviours during the whole day can be assessed [62]. By pretesting, we made adjustments for age and improved the clarity of the questionnaires. However, these child-appropriate questions lead to less detailed data. The advantage of using a parental questionnaire was that more detailed questions could be asked. However, only children’s behaviours outside of school hours can be assessed by them, and the response rate was much lower (66.5%). 

## 8. Conclusions

In the current study, we were able to investigate the effectiveness of the full and partial HPSF compared to control schools regarding children’s health behaviours after 1 and 2 years’ follow-up. Taking all results and limitations into account, we conclude that the full HPSF is effective in promoting children’s health behaviours at T1 and T2 compared with control schools. Focusing on both nutrition and PA components seems to be more effective in promoting healthy behaviours than focusing exclusively on PA. 

## Figures and Tables

**Table 1 nutrients-11-00689-t001:** Children’s PA behaviours: Observed descriptives and estimated 1- and 2-year effects.

		*n*	Full HPSF		Partial HPSF		Control		Full HPSF vs. Control ^b^			Partial HPSF vs. Control ^b^		
			Observed Mean (±SD)	Imp. (%) ^a^	Observed Mean (±SD)	Imp. (%) ^a^	Observed Mean (±SD)	Imp. (%) ^a^	B (95% C.I.)	*p*	ES	B (95% C.I.)	*p*	ES
**Accelerometry**														
**Sedentary (%) ^c^**	**T0**	857	59.7 (±7.19) *		60.9 (±7.02)		61.6 (±6.73)							
	**T1**	929	61.3 (±6.29)	30.1	61.1 (±7.02)	36.5	62.3 (±7.42)	31.1	−0.83 (−1.88–0.23)	0.125	−0.15	0.40 (−0.64–1.45)	0.448	0.07
	**T2**	783	61.5 (±6.84)	28.2	62.3 (±6.92)	21.1	63.4 (±6.99)	21.6	−1.29 (−2.39–−0.19)	**0.021**	−0.23	−0.17 (−1.25–0.90)	0.756	−0.03
**Light PA (%) ^c^**	**T0**	857	32.0 (±5.75) *		31.9 (±5.66) *		31.0 (±5.40)							
	**T1**	929	30.7 (±4.98)	31.1	31.3 (±5.50)	29.4	30.2 (±5.78)	28.5	0.44 (−0.39–1.28)	0.299	0.10	−0.24 (−1.07–0.58)	0.566	−0.06
	**T2**	783	30.6 (±5.20)	21.1	30.1 (±5.44)	17.8	29.3 (±5.30)	17.9	0.94 (0.07–1.81)	**0.034**	0.22	−0.03 (−0.88–0.82)	0.946	−0.01
**MVPA (%) ^c^**	**T0**	857	8.3 (±2.73) *		7.2 (±2.53)		7.4 (±2.58)							
	**T1**	929	8.0 (±2.50)	47.0	7.6 (±2.73)	52.6	7.5 (±2.64)	44.6	0.40 (−0.04–0.85)	0.074	0.17	−0.18 (−0.61–0.26)	0.429	−0.07
	**T2**	783	7.9 (±2.88)	44.4	7.6 (±2.54)	46.1	7.4 (±2.59)	35.8	0.36 (−0.10–0.82)	0.121	0.15	0.19 (−0.26–0.64)	0.407	0.08
**CPM ^c^**	**T0**	857	1209 (±305.3) *		1119 (±274.6)		1117 (±269.5)							
	**T1**	929	1170 (±268.8)	33.9	1137 (±297.7)	39.8	1122 (±311.4)	37.8	26.3 (−18.4–71.1)	0.249	0.11	−19.7 (−64.1–24.8)	0.386	−0.08
	**T2**	783	1162 (±308.4)	30.3	1106 (±274.6)	30.9	1085 (±294.7)	28.4	44.6 (−1.3–90.5)	0.057	0.18	6.8 (−38.2–51.8)	0.767	0.03
**Parent-reported**														
**Total time spent on PA behaviours ** **(sum of min/day)^ d^**	**T0**	743	117.7 (±80.5)		107.1 (±62.8) *		124.0 (±83.1)							
	**T1**	703	109.4 (±68.0)	51.2	109.8 (±59.9)	53.8	130.2 (±82.1)	56.4	−10.7 (−23.6–2.1)	0.104	−0.15	−2.2 (−14.4–10.0)	0.738	−0.03
	**T2**	635	102.1 (±58.9)	42.7	109.0 (±69.2)	54.9	129.2 (±85.1)	45.4	−15.6 (−30.2–−1.0)	**0.037**	−0.22	0.2 (−14.4–14.8)	0.980	0.00
**Total time spent on sedentary behaviours (sum of min/day)^ e^**	**T0**	737	91.3 (±70.9)		93.5 (±73.9)		98.3 (±77.4)							
	**T1**	695	88.0 (±58.5)	46.0	97.4 (±72.7)	35.7	103.4 (±77.5)	35.3	−13.6 (−25.7–−1.6)	**0.026**	−0.21	−3.7 (−15.3–7.6)	0.529	−0.06
	**T2**	617	85.0 (±60.2)	40.5	94.7 (±59.0)	30.8	104.2 (±74.5)	30.2	−18.9 (−31.5–−6.3)	**0.003**	−0.29	−11.1 (−23.6–1.4)	0.085	−0.17

* Significantly different at baseline compared with the control schools, analyzed by ANOVA tests (continuous outcomes). ^a^ Improved: Percentage of children who improved, i.e., changed in a favourable direction, compared with their baseline score. ^b^ Analyzed by linear mixed model analyses (continuous outcomes), adjusted for baseline, gender, study year at T0, SES, ethnicity, and BMI *z*-score at T0. ^c^ Analyzed by linear mixed model analyses (continuous outcomes), adjusted for baseline, gender, study year at T0, SES, ethnicity, BMI *z*-score at T0, mean temperature, sun exposure in hours/day and precipitation in hours/day. ^d^ Included items in total score for PA behaviours: spent time (min/day) in active transport, leisure time PA inside and outside, sports clubs. ^e^ Included items in total score for sedentary behaviours: spent time (min/day) in watching TV, using computer, social media; Bold *p*-value = significant (<0.05) difference; Abbreviations: HPSF = Healthy primary School of the Future; SD = standard deviation; B = coefficient obtained from linear mixed model, indicating a corrected group effect at that specific time point (T1 or T2); C.I. = confidence interval; *p* = *p*-value; ES = Effect size; MVPA = moderate-to-vigorous physical activity; CPM = counts per minute.

**Table 2 nutrients-11-00689-t002:** Children’s dietary behaviours: Observed descriptives and estimated 1- and 2-year effects.

		*n*	Full HPSF		Partial HPSF		Control		Full HPSF vs. Control ^b^			Partial HPSF vs. Control ^b^		
			Observed Mean (±SD)/%	Imp. (%) ^a^	Observed Mean (±SD)/%	Imp. (%) ^a^	Observed Mean (±SD)/%	Imp. (%) ^a^	B (95% C.I.)	*p*	ES	B (95% C.I.)	*p*	ES
**Parent-reported**														
**Healthy dietary behaviours (mean days/week) ^c^**	**T0**	715	5.14 (±1.14)		5.18 (±0.99) *		4.94 (±1.11)							
	**T1**	685	5.44 (±1.00)	61.3	5.40 (±0.94)	56.9	5.06 (±0.99)	48.7	0.20 (0.04–0.37)	**0.015**	0.20	0.07 (−0.09–0.23)	0.405	0.07
	**T2**	593	5.52 (±0.98)	61.3	5.36 (±0.95)	49.2	5.11 (±1.09)	53.8	0.19 (0.01–0.37)	**0.037**	0.19	−0.02 (−0.20–0.16)	0.857	−0.02
**Unhealthy dietary behaviours (mean days/week) ^d^**	**T0**	711	1.09 (±0.62)		1.06 (±0.58)		1.18 (±0.74)							
	**T1**	684	0.97 (±0.65)	56.3	1.00 (±0.62)	49.4	1.19 (±0.72)	38.0	−0.15 (−0.28–−0.03)	**0.017**	−0.23	−0.09 (−0.22–0.03)	0.148	−0.14
	**T2**	590	1.02 (±0.68)	53.8	1.07 (±0.63)	43.1	1.12 (±0.70)	43.4	−0.07 (−0.21–0.07)	0.310	−0.11	0.00 (−0.14–0.15)	0.961	0.01
**Child-reported**														
**Healthy dietary behaviours (0–3) ^e^**	**T0**	781	2.32 (±0.54)		2.27 (±0.57)		2.25 (±0.49)							
	**T1**	1021	2.30 (±0.53)	40.6	2.29 (±0.54)	37.6	2.23 (±0.52)	36.7	0.02 (−0.07–0.10)	0.648	0.04	0.03 (−0.06–0.11)	0.550	0.05
	**T2**	1054	2.30 (±0.50)	45.5	2.25 (±0.53)	44.4	2.21 (±0.54)	46.3	0.02 (−0.07–0.11)	0.637	0.04	0.02 (−0.07–0.11)	0.724	0.03
**Unhealthy dietary behaviours (0–3)^ f^**	**T0**	781	1.12 (±0.56)		1.11 (±0.61)		1.17 (±0.52)							
	**T1**	1021	1.00 (±0.58)	58.5	0.85 (±0.53)	67.0	0.95 (±0.55)	67.0	0.10 (0.02–0.19)	**0.019**	0.20	−0.06 (−0.14–0.03)	0.192	−0.11
	**T2**	1053	1.01 (±0.62)	66.2	0.84 (±0.53)	73.0	0.96 (±0.62)	68.5	0.08 (−0.03–0.18)	0.149	0.14	−0.13 (−0.23–−0.03)	**0.014**	−0.25
**School water consumption ** **(0–3) ^g^**	**T0**	770	1.18 (±1.09)		1.34 (±1.22) *		1.02 (±0.99)							
	**T1**	1014	2.51 (±0.85)	73.3	1.49 (±1.19)	30.1	1.29 (±1.06)	34.6	1.06 (0.86–1.26)	**<0.001**	1.03	−0.13 (−0.33–0.07)	0.203	−0.13
	**T2**	1046	2.59 (±0.80)	76.8	1.54 (±1.17)	40.3	1.26 (±1.15)	40.3	1.17 (0.95–1.38)	**<0.001**	1.14	−0.02 (−0.23–0.20)	0.862	−0.02

* Significantly different at baseline compared to the control schools, analyzed by ANOVA tests (continuous outcomes). ^a^ Improved: Percentage of children who improved, i.e., changed in a favourable direction, compared with their baseline score. ^b^ Analyzed by linear mixed model analyses (continuous outcomes), adjusted for baseline, gender, study year at T0, SES, ethnicity, and BMI *z*-score at T0. ^c^ Included items in total score for healthy dietary behaviours: frequency of consumption of breakfast, fruits, vegetables, water. ^d^ Included items in total score for unhealthy dietary behaviours: frequency of consumption of soft drinks, sports drinks, energy drinks, chocolate, salted snacks, cookies, soft ice-cream. ^e^ Included items in total score for healthy dietary behaviours: frequency of consumption of breakfast, fruits, vegetables, water; range: 0 (never)–3 (every day). ^f^ Included items in total score for unhealthy dietary behaviours: frequency of consumption of soft drinks, sports drinks, energy drinks, chocolate, salted snacks, cookies, soft ice-cream; ^d^ range: 0 (never)–3 (every day). ^g^ School water consumption; range: 0 (never)–3 (daily); Bold *p*-value = significant (<0.05) difference; Abbreviations: HPSF = Healthy primary School of the Future; SD = standard deviation; B = coefficient obtained from linear mixed model, indicating a corrected group effect at that specific time point (T1 or T2), C.I. = confidence interval; *p* = p-value; ES = Effect size.

**Table 3 nutrients-11-00689-t003:** Children’s lunch intake: Observed descriptives and estimated 1- and 2-year effects.

		*n*	Full HPSF		Partial HPSF		Control		Full HPSF vs. Control ^b^		Partial HPSF vs. Control ^b^	
			% Observed	Imp. (%) ^a^	% Observed	Imp. (%) ^a^	% Observed	Imp. (%) ^a^	OR (95% C.I.)	*p*	OR (95% C.I.)	*p*
**Child-reported**												
**Fruit (% yes)**	**T0**	856	37.2		38.2 *		29.9					
	**T1**	1202	62.2	39.8	35.5	13.8	31.3	17.6	2.63 (1.72–4.01)	**<0.001**	0.80 (53–1.22)	0.303
	**T2**	1149	52.6	30.8	42.9	18.2	35.6	24.9	1.41 (0.91–2.17)	0.122	0.88 (0.58–1.35)	0.571
**Vegetables (% yes)**	**T0**	857	23.8		25.4 *		17.8					
	**T1**	1203	53.7	44.8	18.7	8.8	21.2	15.5	3.17 (1.97–5.09)	**<0.001**	0.52 (0.32–0.84)	**0.009**
	**T2**	1145	64.0	46.8	21.5	15.2	23.7	17.8	4.39 (2.68–7.21)	**<0.001**	0.58 (0.35–0.95)	**0.030**
**Grains (% yes)** ^ c^	**T0**	861	93.6 *		89.2 *		80.1					
	**T1**	1208	90.5	5.4	89.4	9.1	86.0	15.4	0.43 (0.22–0.85)	**0.017**	0.68 (0.37–1.23)	0.214
	**T2**	1144	90.1	5.6	88.7	10.0	85.0	15.7	0.45 (0.22–0.91)	**0.027**	0.69 (0.38–1.27)	0.237
**Dairy (% yes)^ d^**	**T0**	852	35.1		40.3		37.0					
	**T1**	1201	73.3	43.6	31.5	13.1	39.1	21.7	4.43 (2.90–6.76)	**<0.001**	0.59 (0.39–0.89)	**0.013**
	**T2**	1146	74.3	45.2	26.9	10.0	39.9	22.8	4.52 (2.93–6.97)	**<0.001**	0.45 (0.29–0.68)	**<0.001**
**Water (% yes)**	**T0**	852	33.9 *		39.6 *		25.9					
	**T1**	1197	55.7	35.8	42.9	18.5	42.2	28.4	1.18 (0.77–1.81)	0.448	0.52 (0.34–0.79)	**0.002**
	**T2**	1143	51.7	34.8	49.6	25.7	35.2	24.2	1.35 (0.87–2.09)	0.185	0.95 (0.62–1.45)	0.810
**Butter (% yes)**	**T0**	862	55.8 *		55.2 *		45.2					
	**T1**	1202	23.1	5.8	48.7	12.5	46.2	17.3	0.22 (0.15–0.33)	**<0.001**	0.74 (0.51–1.06)	0.109
	**T2**	1144	21.1	8.1	47.9	12.9	47.9	18.2	0.19 (0.13–0.30)	**<0.001**	0.64 (0.43–0.94)	**0.024**
**Minimum of two food groups at lunch (% yes) ^e^**	**T0**	869	81.1		85.4 *		77.1					
	**T1**	1210	95.0	17.5	83.7	9.4	80.4	19.1	3.51 (1.88–6.56)	**<0.001**	0.69 (0.41–1.17)	0.171
	**T2**	1151	93.9	19.0	81.1	12.4	80.0	15.6	2.98 (1.59–5.61)	**<0.001**	0.62 (0.36–1.05)	0.078

* Significantly different at baseline compared with the control schools, analyzed by Pearson’s chi-square tests (binary outcomes). ^a^ Improved: Percentage of children who improved, i.e., changed in a favourable direction, compared with their baseline score. ^b^ Analyzed by generalized estimating equations (binary outcomes), adjusted for baseline, gender, study year at T0, SES, ethnicity, and BMI *z*-score at T0. ^c^ Grains consists of the items: bread and cereals. ^d^ Dairy consists of the items: milk/yoghurt and cheese. ^e^ The different food types were: fruits, vegetables, dairy products, grains, water, butter; Bold *p*-value = significant (<0.05) difference; Abbreviations: HPSF = Healthy primary School of the Future; OR = corrected odds ratio obtained from Generalized Estimating Equations (GEE) analysis; C.I. = confidence interval; *p* = *p*-value.

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
