# Peer review of "One- and Two-Year Effects of the Healthy Primary School of the Future on Children’s Dietary and Physical Activity Behaviours: A Quasi-Experimental Study"

_nutrients, 2019, doi:10.3390/nu11030689_

Round 1

Reviewer 1 Report

Review of nutrients-445877

This manuscript addresses an important topic and has the potential to make a worthwhile contribution to the field, particularly given the longitudinal design. However, some issues will need to be addressed before it can be accepted for publication, including providing more context and information about the intervention. Below, I am providing some feedback that the authors may wish to consider in revising their manuscript.

Intervention:

·      My understanding is that the intervention was still in place when T1 and T2 data were collected? It may be useful to clarify this.

·      I suggest that the authors provide some more information about the intervention components. For example, what did the ‘structured PA sessions’ include exactly? How long were they? How were they developed? Did they address any principles related to children’s motivation and behaviour? In discussing their results, the authors seem to ignore the role of the quality of the experience for students.

·      In the introduction, the authors state that the aim of the initiative under investigation is to ‘sustainably integrate health and well-being within the whole school system’. It is unclear, however, what elements of the initiative support sustainability. For example, the fact that external employees implemented the top-down changes with no mentioning of any involvement or training of school staff does not seem to support sustainability. Also, was this a funded project?

·      The authors also state that this initiative is based on the HPS framework. It would be useful to map the initiative on the framework (or give information about the alignment between the two). The framework might also be useful to refer to in the discussion and be considered when discussing the findings.

·      Top-down vs other changes? The authors describe top-down changes in this manuscript but also refer to bottom-up changes or processes, when discussing differences between the full and partial programs. These bottom-up changes seem to be important to describe as they seem to have potentially influenced the results.

Other:

·      Lines 85-87: it may be useful to elaborate on this statement.

·      Line 92: the use of the term ‘mainly’ suggests that the authors expected to see more than just PA changes in the partial HPSF. Why?

·      Line 139: how did the authors treat cases of children with no baseline data?

·      Line 245: which observed participant characteristics?

·      Line 317: is ‘improved’ a more suitable term to use for healthy dietary behaviours?

·      Lines 352-353: I suggest ‘decreased significantly more’

·      Lines 382-384: expand here – why might this be happening?

·      The discussion seems to be missing the opportunity to touch on the so what and uniqueness of the study as well as some key elements such as potential sustainability. The authors may wish to address this in revising their manuscript.

Author Response

We thank the reviewer for the positive feedback regarding the contribution of our paper to the field. We will address all the points of the reviewer point-by-point in the included revision letter.

Reviewer 2 Report

Dear authors,

Your paper is a very interesting study that illustrates the importance of implementing interventions to promote healthy lifestyles to school children, in a new scope. However, some aspects of methodology and results must be further detailed to improve the data provided. For instance, it is necessary to improve and clarify the readability of the methodology and results section, and the discussion about the clinic relevance of their results.

Author Response

We thank the reviewer for the positive feedback regarding the importance and the scope of our paper. We will address all the points of the reviewer point-by-point in the included revision letter.

Round 2

Reviewer 1 Report

Review of nutrients-445877-v2

Thank you for providing clarifications and addressing the feedback provided in the first round of review. I do appreciate the authors’ efforts in this revision. At the same time, I think more careful consideration of the following comments/questions is necessary:

o   Intervention: (How) did the intervention components address any principles related to children’s motivation and behaviour? Were students involved in the bottom-up process? And how did they respond to the intervention component? Again, the authors seem to ignore the role of the quality of the experience for students.

o   Sustainability: the additions/revisions related to sustainability are rather superficial. What does research or relevant frameworks suggest are elements critical to sustainability of school-based interventions? And, how/which were addressed in this project? Related to this, I am unsure why the involvement of external pedagogical employees provided by childcare organisations would change the school’s organisation in a sustainable way? Maybe I am missing something here, but isn’t the involvement of these individuals dependent on funding?

o   Alignment with HPS framework: the authors mention that this framework focuses on a number of elements, including education, school policy, physical and social environment. I would add partnerships, etc. Can the authors provide information (e.g., in a table) in terms of what components each school addressed? For example, it seems that only full HPSF schools addressed education and policy? Further, the authors need to use this information when discussing their results. For example, can differences be attributed to specific elements of HPS used by schools? In other words, what I am suggesting is better integration/articulation of the HPS framework in the manuscript.

o   Line 122: what do the authors mean by ‘school approach’?

Author Response

Dear editor,

Thank you for giving us an extra opportunity to improve our manuscript and re-submit our manuscript. We also want to thank the reviewers for their critical view to help us to further improve the manuscript. Below we explain how we incorporated the comments in our revision. In the manuscript we have used the track changes option. To indicate which changes we made in this second round, we highlighted the new changes in yellow.

Reviewer 2 Report

Dear editor,

The paper entitled: One- and two-year effects of the Healthy Primary School of the Future on children’s dietary and physical activity behaviours, is a very interesting study that illustrates the importance of implementing interventions to promote healthy lifestyles to school children, in a new scope of Healthy Schools. In general, this manuscript is original and it would be of interest to general readers.

Thank you for the information provided. However, I have new questions, and topics not solved.

Author Response

(The authors gave the same response as above.)

Round 3

Reviewer 1 Report

Thank you for providing additional information and clarifications. I have no further comments.

Reviewer 2 Report

Dear,

The paper entitled: One- and two-year effects of the Healthy Primary School of the Future on children’s dietary and physical activity behaviours, is a very interesting study that illustrates the importance of implementing interventions to promote healthy lifestyles to school children, in a new scope of Healthy Schools. In general, this manuscript is original and it would be of interest to general readers.

The authors improve the overall content of the manuscript. There are no more comments.